# Use of Failure Mode and Effects Analysis (FMEA) for Risk Analysis of Drug Use in Patients with Lung Cancer

**DOI:** 10.3390/ijerph192315428

**Published:** 2022-11-22

**Authors:** Shuzhi Lin, Ningsheng Wang, Biqi Ren, Shuang Lei, Bianling Feng

**Affiliations:** 1The Department of Pharmacy Administration and Clinical Pharmacy, School of Pharmacy, Xi’an Jiaotong University, Xi’an 710061, China; 2The Center for Drug Safety and Policy Research, Xi’an Jiaotong University, Xi’an 710061, China

**Keywords:** failure mode and effects analysis, patient medication and post-medication monitoring, SBM-DEA, lung cancer

## Abstract

It is crucial to investigate the risk factors inherent in the medication process for cancer patients since improper antineoplastic drug use frequently has serious consequences. As a result, the Severity, Occurrence, and Detection rate of each potential failure mode in the drug administration process for patients with lung cancer were scored using the Failure Mode and Effect Analysis (FMEA) model in this study. Then, the risk level of each failure mode and the direction of improvement were investigated using the Slacks-based measure data envelopment analysis (SBM-DEA) model. According to the findings, the medicine administration process for lung cancer patients could be classified into five links, with a total of 60 failure modes. The risk of failure modes for patient medication and post-medication monitoring ranked highly, with unauthorized use of traditional Chinese medicine and folk prescription and unauthorized drug addition (incorrect self-medication) ranking first (1/60); doctor prescription was also prone to errors. The study advises actively looking at ways to decrease the occurrence and difficulty of failure mode detection to continually enhance patient safety when using medications.

## 1. Introduction

Globally, the estimated annual cost associated with medication errors is $42 billion [1]. The World Health Organization has indicated that unsafe medication practices and medication errors are leading causes of injury and avoidable harm in healthcare systems throughout the world [2]. Unsafe medication practices and medication errors increase the risk of many types of adverse reactions and may even be life-threatening. Therefore, how to ensure safe drug use and improve the quality of medical services is a major healthcare problem worldwide.

Chemotherapeutics are the second most common drug type associated with fatal drug errors [3]. An evaluation report of United States poison centers also showed that chemotherapeutics are among the most common drugs associated with medication errors [4]. Cancer treatment is a very complex process. The vulnerability of these patients’ physiological state, the toxicity of antitumor treatment, and the complexity of the nursing process [5] make patients with cancer face higher risks in the treatment process. The medication process is not only an essential link in the administration of cancer treatment but is also associated with high medical risk. Lung cancer has the highest mortality rate and the second highest incidence rate in the world, so the safety of its administration deserves more attention [6].

As a risk management tool, failure mode and effects analysis (FMEA) can proactively identify potential faults and hidden dangers in a system or process, analyze their causes and effects, and propose preventive measures to reduce the occurrence rate and severity of faults [7,8]. FMEA technology began to be applied to the American aerospace industry in the mid-1960s and entered the field of medical devices in the late 1970s. In recent years, FMEA technology has been widely used in different medical environments, such as the healthcare process, hospital management, hospital informatization, and medical equipment and production [7]. The advantage of FMEA technology lies in “early prevention” rather than “post remedy,” thus indicating how to invest limited resources to achieve greater healthcare improvement. The Joint Commission on Accreditation of Healthcare Organizations requires qualified hospitals to regularly carry out forward-looking assessments using FMEA [9,10]. China’s newly revised Good Manufacturing Practice of Medical Products also emphasizes the importance of risk assessment in the process of drug production [11].

From the perspective of nursing, Ashley [12] identified 30 failure modes of chemotherapy administration with the help of FMEA, including confusion of patients during administration, omission of drugs, and incorrect doses. The author thus formulated relevant remedial strategies for the possible causes of high-risk failure modes. Anjalee et al. [10] used FMEA to identify failure modes, causes, and effects in the dispensing process at a tertiary care hospital. Weber et al. [13] used FMEA to analyze the security vulnerabilities in the process of chemotherapy prescription, compounding, transport, and administration and proposed corresponding measures to improve the safety of patients in the process of chemotherapy. FMEA is also used to study problems in prescribing, dispensing, and administration of children’s chemotherapy [14]. Previous studies have shown that FMEA is a useful tool to proactively evaluate the safety of any step in drug use and formulate corrective measures.

At present, FMEA of cancer chemotherapy drugs is based on healthcare professionals’ examination of the potential failure modes that affect safe drug use in the stages of prescription, dispensing, and administration. The effect of patients’ own factors on safe drug use has not been considered. Weingart [5] pointed out that chemotherapy medication errors occur in all stages of the drug use process and that patient medication is an important part. A previous study [15] also revealed that lung cancer patients in Shaanxi Province lacked awareness of medication safety. The National Coordinating Council for Medication Error Reporting and Prevention defines a medication error as any preventable event that may cause or lead to inappropriate medication use or patient harm while the medication is in the control of the healthcare professional, patient, or consumer [10]. Such events may be related to professional practice, healthcare products, procedures, or systems, including prescribing, order communication, product labeling, packaging, nomenclature, compounding, dispensing, distribution, administration, education, monitoring, and use.

Therefore, the present study innovatively incorporated patient medication and post-medication monitoring into the medication process. FMEA was conducted on the whole process of lung cancer medication, including doctor prescription, pharmacist deployment, nurse administration, patient medication and post-medication monitoring, and medical organization management; the failure modes of each link were explored; the risk levels of each failure mode were ranked.

## 2. Materials and Methods

This cross-sectional study was conducted in Shaanxi Province (a representative province in western China) from June 2021 to August 2021. We assessed the geographical environment and economic development of various Shaanxi Province regions. The province is divided into three regions (Northern, Central and Southern Shaanxi) and one provincial capital city (Xi’an). Using the median per capita GDP of 11 regions in Shaanxi Province as a benchmark, in addition to Xi’an, Yulin (120,900 RMB, high per capita GDP), Hanzhong (45,000 RMB, medium per capita GDP), and Weinan (34,500 RMB, low per capita GDP) were chosen as study cities from each of the three regions of Northern Shaanxi, Southern Shaanxi, and Central Shaanxi.


**Step 1: Planning and preparation: build initial medication process and potential failure modes**


Prior to the formal discussion, to sort out the components involved in the medication process and summarize preliminary potential failure modes, the researchers read the literature [5,16,17,18,19,20,21,22,23,24,25,26] on factors involved in medication errors, healthcare professionals’ perceptions of medication safety, patient involvement in medication safety content, and a study [15] of medication safety awareness of patients with lung cancer in Shaanxi Province. Because there is no standard against which to measure whether the initially constructed failure modes were comprehensive, we focused on the time at which the data were saturated; that is, an increase in the number of articles read will not produce new failure modes.


**Step 2: Set up a team for FMEA**


Considering the accessibility of medical resources and the doctors’ medical levels, we limited the selection of team members to cancer hospitals and hospitals with oncology or respiratory departments in Xi’an, Shaanxi Province (provincial capital city). We selected doctors and head nurses with more than 10 years of working experience and technical titles of vice-senior title or above and clinical pharmacists specializing in the treatment of lung cancer. These individuals were highly professional and representative in the clinical first-line treatment of lung cancer. The multidisciplinary FMEA team in this study comprised 11 healthcare professionals (four doctors, three nurses, and four pharmacists).


**Step 3: Modify the preliminary model of Step 1 through interviews and obtain the final version of the medication process and potential failure modes**


We were unable to hold a discussion in the form of a symposium because of the constraints of this study; instead, we performed face-to-face interviews between the researcher and each member of the FMEA team. For each semi-structured interview, the interview guidelines were based on the pre-prepared medication process and preliminary potential failure modes. Each interview lasted more than 30 min, and all interviews were recorded and transcribed word for word.

The interview content mainly included the following questions: Which processes and failure modes are not included and which are unnecessary? What can be merged? What needs further refinement? After the interview, the researchers revised the questions and suggestions raised during the interview from the perspectives of content professionalism and logical rationality. The revised catalog of the medication process and potential failure modes for patients with lung cancer was sent to the team members for final consensus.


**Step 4: Score the Severity (S), Occurrence (O), and Detection (D) of each failure mode in the form of a questionnaire**


The questionnaire was drawn. The content of the questionnaire included two parts (see Appendix A for details): (1) demographic information, including sex, age, educational background, work, working years, professional title, working place, working hours, and related factors, and (2) the modified and improved failure modes in Step 3. Each failure mode was scored from the three dimensions of consequence (S, O, and D), and the score for each ranged from 1 to 10. The scoring criteria [27] are shown in Table 1.

The questionnaires were distributed to cancer hospitals and hospitals with oncology or respiratory departments in Yulin, Hanzhong, Weinan, and Xi’an. To ensure the professionalism of the rater, the medical personnel who filled in the questionnaire were required to be professionals engaged in front-line treatment of lung cancer in the hospital (doctors, nurses, and pharmacists with more than 10 years of working experience or staff members with technical titles of vice-senior title and above). The distribution of the questionnaire included filling out an online version of the questionnaire and filling out a paper version of the questionnaire on-site. The exclusion criteria for the questionnaire were as follows: (1) the questionnaire took less than 500 s to answer (the researcher found that the submission time was about 400 s if the respondent only randomly checked the questionnaire options without carefully answering each item), (2) the rater of the questionnaire did not meet the requirements, (3) the responses to the questionnaire were similar, and (4) more than five questions in a row were checked with the same option.

After the questionnaires were collected, the average score of the S, O, and D dimensions of each failure mode was calculated.


**Step 5: Use of slacks-based measure data envelopment analysis (SBM-DEA) model to sort failure modes**


The risk priority number (RPN) in FMEA is calculated as RPN = S × O × D. The S, O, and D scores are subjective and have no distance measure and multiplying them is an invalid method of calculation. In 1998, Bowles [28] pointed out that the severity and detection scale of FMEA is qualitative; for example, a score of 6 is not twice as severe as a score of 3. Another limitation of FMEA is that the S, O, and D dimensions have the same importance, and different combinations of S, O, and D may produce the same RPN. Studies have shown that DEA can be used to improve the detection ability of FMEA [28,29,30]. DEA is used to evaluate the relative effectiveness of comparable units by means of linear programming according to multiple input and output indicators. DEA does not need to assume the weight; it obtains the optimal weight according to the input-output data of the decision-making unit, excluding the influence of subjective factors. DEA can sort failure modes from the perspective of efficiency, and the relaxation variables generated can provide quantitative information for improving strategies.

The failure modes generated in FMEA can be regarded as the decision-making units in DEA, and S, O, and D can be regarded as the input index in DEA. We set the output index to 1 and adopted the SBM-DEA model of input angle. The programming model is described below:$$\begin{array}{c} (\mathrm{SBM-I})\;\mathrm{MIN}\;\rho =1-\frac{1}{m}\Sigma_{i=1}^{m}x_{io}/x_{io}\\ subject\;to\\ x_{io}=\sum_{j=1}^{n}\lambda_{j}x_{ij}+s_{i}^{-}\\ y_{ro}\leq \sum_{j=1}^{n}\lambda_{j}y_{rj}\\ \;s_{i}^{-}\geq 0,\;\lambda_{j}\geq 0,\\ i=1,\ldots ,m\\ r=1,\ldots ,s\\ j=1,\ldots ,d \end{array}$$

## 3. Results

### 3.1. Medication Process and Failure Mode

By summarizing the literature and combining the preliminary investigation, we divided the drug use process for patients with lung cancer into five links: doctor prescription, pharmacist deployment, nurse administration, patient medication and post-medication monitoring, and medical organization management. The division of the process was unanimously recognized by team members during the interview.

In the interview, we deleted the failure mode of “Handwritten prescription, the prescription is difficult to identify”, mainly because the popularity of electronic prescription means that handwritten prescription almost never occurs in clinical practice. “History of previous Adverse Drug Reaction” is a key consideration for doctors when prescribing, and it was deleted in the process of patient medication and post-medication monitoring. The following failure modes were added: Doctor prescription: Not specifying the standard and reasonable chemotherapy administration sequence, not giving reasonable chemotherapy pretreatment steps to patients; patient medication and post-medication monitoring: Unauthorized use of traditional Chinese medicine and folk prescription and unauthorized drug addition (incorrect self-medication); nurse administration: Improper drug delivery interval, incorrect drug delivery method, chemotherapy drug infusion duration (infusion speed) not controlled within the standard time range; medical organization management: hospital informatization is not high and there is no pre-examination system. Some failure modes were combined: “Improper drug selection” and “Drug selection without indications” were combined into one failure mode. To refine the failure mode, “Off-label drug use” was changed to “Incorrect off-label drug use” in consideration of the global recognition of off-label drug use and the need for precise and individualized treatment for cancers.

Finally, we divided the drug administration process into five links and summarized and refined the potential failure modes in each link. There were 60 potential failure modes in total (see Appendix A for details): doctor prescription (*n* = 13), pharmacist deployment (*n* = 8), nurse administration (*n* = 12), patient medication and post-medication monitoring (*n* = 16), and medical organization management (*n* = 11).

### 3.2. Demographic Information of the Questionnaire

Fifty questionnaires were collected, and 29 valid questionnaires were obtained after the application of the exclusion criteria (effective recovery of 58.00%).

The demographic information in the 29 valid questionnaires showed that the proportion of male and female raters was similar (see Table 2 for details). The raters’ average age was 41.9 years. Most raters were aged 31 to 40 years (48.28%), followed by 41 to 50 years (41.38%). Bachelor’s degrees accounted for 55.17% of the total, and doctoral degrees accounted for 17.24%. The proportion of doctors and nurses was 48.28% and 37.93%, respectively; pharmacists accounted for 13.79%. The raters’ average number of working years was 18.6, and most had worked for 10 to 30 years (93.10%). The rate of vice-senior title and above was 41.38%; 55.17% of raters worked in Xi’an, 13.79% in Weinan, 20.69% in Hanzhong, and 10.34% in Yulin. Additionally, 58.62% of respondents worked 40 to 50 h per week, whereas 10.34% worked more than 60 h per week.

### 3.3. Risk Ranking of Failure Modes

The results of the SBM-DEA model were sorted in ascending order according to efficiency score. A lower efficiency score of the failure mode was associated with a higher risk priority (see Table 3 for details). The failure mode with the highest risk was located in the link of medication and post-medication monitoring: Unauthorized use of Traditional Chinese medicine and folk prescription and unauthorized drug addition (incorrect self-medication) (No. 38), with a score of 0.6285. The lowest risk was found in the seven failure modes with an efficiency score of 1, which were doctor prescription: prescription missing drugs (No. 8); medical organization management: the department does not have an environment for chemotherapy drug preparation, the environment of infusion drug dispensing is poor, the PIVAS is missing or in poor condition (No. 60), the pharmacy has deteriorated and expired drugs (No. 59), and the reporting system of adverse drug reactions/adverse events is incomplete and the reporting system is missing (No. 54); pharmacist deployment: pharmacists are far away from clinical practice, exhibit poor participation in ward rounds, and fail to provide pharmaceutical care such as medication consultation and post-medication monitoring to patients in a timely manner (No. 21); additionally, when patients make enquiries after taking drugs, pharmaceutical care is not provided in place, medication consultation service is missing (No. 20), and drugs are missed (No. 15).

The sorted failure modes were divided into quartiles (zones Q1–Q4). Zone Q1 contained the top 25% failure modes according to risk grade, which was also in urgent need of improvement. All failure modes (nine in total), which accounted for 56.25% of patient medication and post-medication monitoring, were located in Q1. Q1 contained four drug use links, excluding the link to medical organization management. Figure 1 shows various failure modes according to risk (in order from high to low). The color distribution of each link shows that patient medication and post-medication monitoring had the highest risk, followed in order by doctor prescription, nurse administration, pharmacist deployment, and medical organization management.

The failure mode with the highest risk in doctor prescription ranked second in risk level: prescribing drugs with incompatibility and drug interactions (incompatibility and drug interactions are unknown) (No. 9). The failure mode with the highest risk in pharmacist deployment was no review of the doctor’s prescription and prescription error not detected in time (No. 17), ranking 13th. Nurse administration in the highest risk of failure mode was ranked third, namely when equipped with drugs is interrupted, interference, distractions, etc. (No. 24). Failure modes with the highest risk in medical organization management ranked relatively low, at 24th: insufficient continuing education and drug safety knowledge training, failure to become updated in the latest knowledge in the field, and lack of training on relevant operating procedures (No. 51).

The SBM-DEA model can not only obtain the efficiency scores of each failure mode but can also provide radial movement for analyzing and judging the gap between each input index of the failure mode and the ideal score (efficiency score of 1), providing a quantitative reference for formulating improvement strategies (see Appendix B for details). Among the improvement rates of the S, O, and D dimensions, the least change occurred in the degree of S. Only two failure modes had a change rate of more than 20%, and 56.67% of failure modes had a change of 0%. O and D had significant changes, and failure modes with a change rate of more than 50% were found in both of them. Occurrence modes with a change rate of more than 30% accounted for 43.33%, and failure modes with a change rate of more than 30% accounted for 46.67%.

## 4. Discussion

In this study, FMEA was used to divide the whole drug administration process for patients with lung cancer in Shaanxi Province, China into five links and 60 failure modes. Among the drug administration links, patient medication and post-medication monitoring were rarely included, and SBM-DEA was used to rank the risk of failure modes. The links with the most failure modes and the greatest risk in drug use in patients with lung cancer in Shaanxi Province of China were patient medication and post-medication monitoring. This finding is consistent with the conclusion that patients with lung cancer had poor medication safety awareness in our previous study.

The World Alliance for Patient Safety states that patients should become active participants in medical safety rather than passive recipients [31]. Existing studies suggest that patients and their families have been regarded as so-called “vigilant partners” in safe cancer care [21]. Patients generally have a positive attitude toward their participation in the practice of safe drug use [32,33], but their evaluation of their own participation in the process of error prevention is less positive [34]. Patients’ intention to participate is affected by clinical diseases and related to attitude, willpower, and other factors [35]. Observation of the failure modes of patient medication and post-medication monitoring shows that patients have certain deficiencies in their understanding of drugs. In addition, in a retrospective study, oncology nurses found that it is not uncommon for patients to realize medication mistakes but fail to convey their observation results for various reasons [36]. Anjalee suggested [10] the creation of a patient consultation platform to highlight the significance of patient–provider communication while using the FMEA approach to analyze safety in the dispensing process. Medical staff including physicians, pharmacists, and nurses need to develop different strategies to educate patients about safe drug use and advocate active participation of patients in their own safe drug use. Such strategies may include preventing self-medication without authorization, learning the correct storage and identification of drugs, learning to read drug instructions, giving attention to adverse drug reactions during medication, and providing encouragement and support to patients [35,36]. This can reduce the frequency of events to a certain extent, which is in line with the direction of improvement in failure modes.

The second risk level is the link to doctor prescriptions, which is also very prone to errors. Previous studies using the FMEA model also found [13] that key steps in the administration of intravenous oncology chemotherapy are associated with prescribing, and a study of the safety of chemotherapy in children had shown [14] that prescribing is an area of high RPN. Patients with cancer usually have more drug combinations and comorbidities and are more sensitive to drug interactions, contraindications, and dosage-related and other errors. Although tools such as computerized physician order entry systems are effective in intercepting some prescribing errors, 30% to 40% of prescribing errors are unavoidable [37]. Since 2018, many hospitals in China have been equipped with an “information-based pre-prescription review system” to quickly review in real time the rationality of drug use, advance the original post-prescription review process, find problematic prescriptions from the source, and reduce drug use errors in the prescription stage [38,39]. Therefore, it is necessary to establish digital hospitals using computers and other sciences and technologies, which can improve the detection of medication errors, reduce the occurrence of medication errors, and ensure the safety of patients.

Nurses are the executors of drug administration and the final gatekeepers of the drug administration process. Thus, the concentration and seriousness of nurses’ work directly affect patient safety. Ashley’s study [12] showed that the main causes of medication administration risks include lack of knowledge and experience of caregivers, lack of control measures for medication risks, and interruptions and distractions in clinical work. Our findings also confirm the risk of disruption to the dispensing process and the persistent need to explore ways to manage interruptions and distractions during nurse administration [26]. Additionally, we can monitor nurses’ medication administration behavior through nurse leaders and respond quickly to medication errors, and nurses must be clear that medication safety is a high-priority strategic goal [40]. The responsibilities of pharmacists include monitoring physicians’ prescriptions for drug interactions and providing patients with drug knowledge; however, the risk ranking of failure modes corresponding to the responsibilities of pharmacists is still high. It is, therefore, important to clarify the rights and obligations of pharmacists and strengthen the construction of the team of pharmacists, which is also in line with the national conditions of the relatively late development of pharmacists in China [41]. Although the risk in medical organization management is low, hospital management, as the establisher of rules and regulations, should establish more professional and detailed rules and regulations on medication safety, increase continuing education and training on medication safety, keep up to date with the latest knowledge in the field, and organize training on relevant operating procedures. Moreover, the relationships among medical staff such as doctors, pharmacists, and nurses should be coordinated, and communication among medical staff members should be enhanced.

The main strength of this study is that it is the first to include patient medication and post-medication monitoring in the medication process, and FMEA was conducted on the whole medication process for patients with lung cancer. Additionally, the SBM-DEA model was adopted to deal with the risk ranking of each failure mode, which overcame the limitations of the RPN to a certain extent and provided a quantitative direction for the improvement of the failure mode. The results of our study may serve as a reference for the formulation of drug safety measures for patients with lung cancer.

Because patients do not have sufficient expertise, we did not include patients in our FMEA team, which was composed entirely of medical staff. Medical staff may pay more attention to the failure modes of patient medication and post-medication monitoring. However, some medical staff, as hospital stakeholders, may not make many negative comments on the organizational management segment of the healthcare organization, which may bias the results somewhat. In addition, we did not implement the process of applying the results to clinical practice. In future work, we may further develop in-hospital interventions for failure modes and assess changes in risk before and after the intervention.

## 5. Conclusions

FMEA can identify gaps in the medication administration process for patients with lung cancer that require urgent improvement and help to propose appropriate improvement measures. This study showed that patient medication and post-medication monitoring play an important role in the whole process of medication administration. It is essential to promote the active participation of patients in their own safe medication administration and raise their awareness of safe medication administration.

## Figures and Tables

**Figure 1 ijerph-19-15428-f001:**
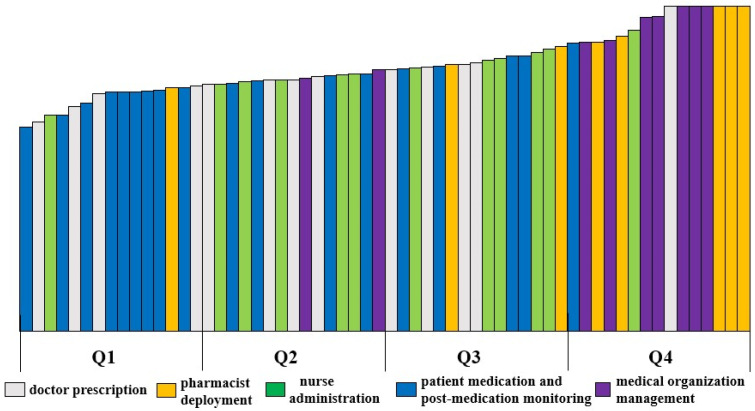
Histogram of each failure mode in order of risk.

**Table 1 ijerph-19-15428-t001:** FMEA scoring criteria.

Severity (S)		Occurrence (O)		Detection (D)	
1	No impact	1	Unlikely to happen	1	Always monitor for such risks and errors in advance
2	Slight impact	2,3,4	Low likelihood	2,3	Likely to be monitored
3	Causes moderate problems and affects patients	5,6	Moderate	4,5,6	Moderately likely to be monitored
4,5	Causing major problems, affecting the patient	7,8	High	7,8	Less likely
6,7	Causing minor harm to the patient	9,10	Very high	9,10	Virtually impossible to detect such risks and errors in advance
8,9	Causing significant harm to the patient				
10	Causing death to the patient				

**Table 2 ijerph-19-15428-t002:** Basic information of raters.

Demographic Information		Number of People	Percentage of (%)
Sex	Male	15	51.72%
	Female	14	48.28%
Age (years)	<30	0	0.00%
	31–40	14	48.28%
	41–50	12	41.38%
	>50	3	10.34%
Academic degree	Bachelor’s degree	16	55.17%
	Master degree	8	27.59%
	Doctoral degree	5	17.24%
Job Title	Doctor	14	48.28%
	Pharmacist	4	13.79%
	Nurse	11	37.93%
Years of work (years)	10–20	17	58.62%
	21–30	10	34.48%
	>30	2	6.90%
The position is vice-senior title and above	Yes	12	41.38%
	No	17	58.62%
Workplace	Xi’an	16	55.17%
	Weinan	4	13.79%
	Hanzhong	6	20.69%
	Yulin	3	10.34%
Hours of work (hours/week)	<40	1	3.45%
	40–50	17	58.62%
	51–60	8	27.59%
	>60	3	10.34%

**Table 3 ijerph-19-15428-t003:** Scores of S, O, and D dimensions of failure mode and efficiency score of SBM-DEA model.

Link	Number	S	O	D	Score
D	NO. 38	6.45	5.00	5.24	0.6285
A	NO. 9	6.72	4.17	5.45	0.6436
C	NO. 24	6.14	4.66	5.00	0.6644
D	NO. 39	6.38	4.79	4.62	0.6646
A	NO. 10	6.62	3.69	4.83	0.6907
D	NO. 41	5.24	4.66	5.10	0.7021
A	NO. 4	6.72	3.86	3.66	0.7303
D	NO. 44	4.69	5.00	4.79	0.7347
D	NO. 42	4.72	5.03	4.69	0.7358
D	NO. 43	5.10	4.48	4.69	0.7360
D	NO. 48	5.28	5.24	3.93	0.7379
D	NO. 40	4.79	5.03	4.48	0.7415
B	NO. 17	6.24	3.93	3.76	0.7480
D	NO. 46	5.21	4.72	4.14	0.7488
A	NO. 2	5.69	3.72	4.55	0.7539
A	NO. 6	4.76	4.41	4.72	0.7592
C	NO. 31	7.07	3.14	3.62	0.7607
D	NO. 49	6.72	3.24	3.72	0.7627
C	NO. 26	6.69	3.24	3.66	0.7684
D	NO. 37	6.45	2.97	4.28	0.7696
A	NO. 1	6.24	3.45	3.79	0.7731
C	NO. 25	6.21	3.59	3.69	0.7731
A	NO. 12	6.00	3.38	4.17	0.7739
E	NO. 51	5.31	4.14	4.03	0.7789
A	NO. 3	6.28	3.31	3.72	0.7839
D	NO. 36	5.38	3.66	4.38	0.7854
C	NO. 22	6.72	3.34	3.24	0.7878
C	NO. 32	4.76	3.97	4.59	0.7909
D	NO. 45	5.31	3.90	4.07	0.7913
E	NO. 57	5.21	5.10	3.24	0.8029
A	NO. 13	4.93	3.83	4.31	0.8033
D	NO. 35	5.52	3.38	4.17	0.8062
C	NO. 30	4.52	3.90	4.66	0.8088
A	NO. 5	5.62	4.07	3.34	0.8110
D	NO. 47	5.90	3.31	3.66	0.8154
B	NO. 16	7.10	2.97	3.00	0.8189
A	NO. 7	5.31	3.48	4.03	0.8201
A	NO. 11	5.00	3.90	3.83	0.8252
C	NO. 23	5.76	2.90	4.14	0.8323
C	NO. 27	4.90	3.66	3.97	0.8386
E	NO. 58	5.86	3.34	3.24	0.8460
E	NO. 53	5.17	4.07	3.28	0.8476
C	NO. 33	4.62	3.45	4.21	0.8578
C	NO. 28	6.10	3.07	3.03	0.8664
B	NO. 14	7.14	2.76	2.62	0.8748
D	NO. 34	5.28	3.07	3.59	0.8860
E	NO. 52	4.90	3.83	3.21	0.8876
B	NO. 19	4.45	4.07	3.69	0.8889
E	NO. 56	4.52	3.86	3.41	0.8927
B	NO. 18	6.48	2.24	3.24	0.9080
C	NO. 29	5.31	2.86	3.31	0.9261
E	NO. 55	4.55	3.03	3.34	0.9653
E	NO. 50	4.72	3.03	3.17	0.9683
E	NO. 59	6.45	2.21	2.41	1.0000
B	NO. 15	5.24	3.14	2.55	1.0000
E	NO. 60	5.21	4.14	2.34	1.0000
E	NO. 54	4.52	2.93	3.14	1.0000
A	NO. 8	4.31	3.38	3.79	1.0000
B	NO. 20	4.21	3.90	3.90	1.0000
B	NO. 21	4.21	4.52	3.69	1.0000

Note: A represents doctor prescription, B represents pharmacist deployment, C represents nurse administration, D represents patient medication and post-medication monitoring, and E represents medical organization management.

## Data Availability

The data that support the findings of this study are available from the corresponding author upon reasonable request.

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
