# Peer review of "Use of Failure Mode and Effects Analysis (FMEA) for Risk Analysis of Drug Use in Patients with Lung Cancer"

_ijerph, 2022, doi:10.3390/ijerph192315428_

Round 1

Reviewer 1 Report

Very interesting manuscript, I do truly congratulate you for doing it. However, I do have a few comments:

First: I do believe the abstract need to be rewritten and clarified. I was not able from reading the abstract at the first time to understand the method and results clearly. Also, try to avoid using "We" in scientific writing. 

In the Literature review, you mention in line 48 previous studies, while you only mentioned one study in the previous sentence. I do believe the rest of the paragraphs need to be reorganized and linked to each other. I felt you jumped between the second three paragraphs without linking them together.

In the methods section, I think you need to show exactly how you selected the cities rather than citing your previous papers. Also, In line 99, you mentioned that you read a large amount of literature, I do believe you need to delete this sentence and mention exactly what article you used to create the survey. 

In the results: please do not use appreciation in the title.

In the discussion, I would like to compare your findings to similar studies using the same methods. 

Great manuscript!

Good Luck!

Reviewer 2 Report

Lin S et al have tried to their best to investigate factors for Risk Analysis of Drug Use in Patients with Lung Cancer, using Failure Mode and Effects Analysis (FMEA). Overall, the paper looks well written and presented, although there are few mistypos errors and language to be corrected.

All sections meet basic requirement for scientific paper, and are full of important content. Nevertheless, i would like to stress out on the method section. The authors declared: "To ensure the professionalism of the rater, the medical personnel who filled in the questionnaire were required to be professionals engaged in front-line treatment of lung cancer in the hospital (doctors, nurses, and pharmacists with more than 10 years of working experience or staff members with technical titles of vice-senior title and above).

Questions:

Q1: Oncology Pharmacist (Oncology Nurse is almost in its genesis stage as individual field) is China is quite a new specialized area of work as most of pharmacists working in oncology had done it just through general training. So how did authors confirm that the 4 clinical pharmacist were specialized oncology pharmacist with a working experience of more than 10 years in Oncology?

Q2: Please do sub-category professional of age and working years experience to better provide sound findings regarding their impact in the questionnaires responses. This is due to the fact that even nurse, most of them are general and dispatched across the hospital based on the Medical Staffing demand.
